# Task Assignment and Path Planning Mechanism Based on Grade-Matching Degree and Task Similarity in Participatory Crowdsensing

**DOI:** 10.3390/s24020651

**Published:** 2024-01-19

**Authors:** Xiaoxue He, Yubo Wang, Xu Zhao, Tiancong Huang, Yantao Yu

**Affiliations:** 1School of Microelectronics and Communication Engineering, Chongqing University, Chongqing 400044, China; 202112131098@stu.cqu.edu.cn (X.H.); htc@cqu.edu.cn (T.H.); 2Beijing Smart-Chip Microelectronics Technology Co., Ltd., Beijing 100192, China; wangyubo@sgchip.sgcc.com.cn (Y.W.); zhaoxu@sgchip.sgcc.com.cn (X.Z.)

**Keywords:** participatory crowdsensing (PCS), grade-matching degree and similarity-based mechanism (GSBM), multi-task assignment (MTA), task similarity, grade-matching degree, improved ant colony optimization (IACO) algorithm

## Abstract

Participatory crowdsensing (PCS) is an innovative data sensing paradigm that leverages the sensors carried in mobile devices to collect large-scale environmental information and personal behavioral data with the user’s participation. In PCS, task assignment and path planning pose complex challenges. Previous studies have only focused on the assignment of individual tasks, neglecting or overlooking the associations between tasks. In practice, users often tend to execute similar tasks when choosing assignments. Additionally, users frequently engage in tasks that do not match their abilities, leading to poor task quality or resource wastage. This paper introduces a multi-task assignment and path-planning problem (MTAPP), which defines utility as the ratio of a user’s profit to the time spent on task execution. The optimization goal of MATPP is to maximize the utility of all users in the context of task assignment, allocate a set of task locations to a group of workers, and generate execution paths. To solve the MATPP, this study proposes a grade-matching degree and similarity-based mechanism (GSBM) in which the grade-matching degree determines the user’s income. It also establishes a mathematical model, based on similarity, to investigate the impact of task similarity on user task completion. Finally, an improved ant colony optimization (IACO) algorithm, combining the ant colony and greedy algorithms, is employed to maximize total utility. The simulation results demonstrate its superior performance in terms of task coverage, average task completion rate, user profits, and task assignment rationality compared to other algorithms.

## 1. Introduction

Recently, the mobile crowd-sensing (MCS) paradigm has attracted attention from many researchers. The main advantage of MCS is that the deployment of several static sensors over a large geographical region is reduced and then replaced by willing users with the required sensing equipment in their smart devices [1,2]. Indeed, users may now collect diverse observations about the physical world during their journeys, using the sensors integrated into their smartphones. However, the anticipated level of engagement from the crowd in such sensing tasks varies [3,4]. This contribution can take two forms: (i) passive involvement in the background, referred to as opportunistic crowdsensing [3,5], or (ii) proactive participation, known as participatory crowdsensing [6,7].

Opportunistic crowdsensing makes it easy to gather a multitude of observations spanning across time and space. Indeed, users simply install a mobile application on their smartphones. However, opportunistic crowdsensing cannot guarantee that the collectively sourced sensor data will yield sufficiently accurate knowledge. Participatory crowdsensing (PCS) has the potential to bring observations of higher quality since the end-user is conscious of carrying out a sensing task. Furthermore, supporting incentive mechanisms allow assigning tasks to the most effective crowd [8], while complementary path-planning schemes allow for maximizing the number of tasks that each participant may achieve [9]. In recent years, PCS has permeated a broad range of applications, such as checking traffic conditions [10], building smart cities [10], detecting air quality [11], etc.

A common challenge of PCS is to achieve optimal task assignment and path planning. Considerable efforts have been devoted to task assignment and path planning mechanisms that rationally assign tasks to users, based on various metrics such as profit [12,13,14], space and time [15,16,17], and task coverage [18,19]. Despite the considerable body of research devoted to optimizing PCS systems, notable lacunae persist in the domains of task assignment and path planning.

Existing studies primarily focus on the assignment of a single task, ignoring the influence of task-to-task association characteristics. In practical scenarios, users often exhibit a preference for performing similar tasks, as this contributes to heightened efficiency and reduces the time cost associated with task completion. However, the existing research [20] has inadequately addressed the consideration of task similarity in task assignment, constraining the potential of PCS systems to enhance task completion rates.

In reality, users may perform tasks that correspond to their abilities. For instance, low-capability users may attempt high-difficulty tasks to obtain greater rewards, resulting in suboptimal task completion quality. High-capability users may complete multiple easier tasks to reduce the risk of failure and ensure a more stable task completion rate. However, this can lead to the wastage of system resources and an imbalance in task allocation. Previous researchers attempted to address the issue, such as by providing additional rewards or incentives for completing difficult tasks [21]. However, the implementation of these measures will increase the cost, limiting the incentive effect. 

As the task assignment and path-planning process is a dynamic combinatorial optimization problem, this paper proposes to use a multi-task assignment and path-planning problem (MTAPP). In order to solve this problem, this paper proposes a grade-matching degree and similarity-based mechanism (GSBM) in a PCS. An improved ant colony optimization algorithm (IACO) has been adopted to effectively assign appropriate tasks to mobile users and plan suitable paths. The main contributions are as follows:We propose the MTAPP, which defines the ratio of each user’s profit to the time needed for completing the task as the utility of the user in terms of task allocation and path planning. The goal of optimization is to maximize the utility of all users in task assignment.We propose a grade-matching degree pricing mechanism in the GSBM, where the degree of matching determines a portion of the expected reward.We establish a mathematical model based on similarity in the GSBM, delving into research on the impact of task similarity on user task completion.We adopt an improved ant colony algorithm (IACO) to maximize the utility of all users, in terms of task assignment; it efficiently assigns suitable tasks to the corresponding mobile users and plans optimal paths.We evaluate the performance of GSBM through extensive experiments. The experimental results demonstrate the promising efficiency of the proposed approaches in maximizing the assignment rationality.

The remainder of this paper is structured as follows. In Section 2, the related works in this field are reviewed. In Section 3, the utility-based task allocation problem is introduced. In Section 4, the GSBM system model is introduced. We also provide a detailed introduction to the optimized ant colony algorithm, and, in Section 5, the performance of the GSBM is evaluated. In Section 6, we present a summary of the research. 

## 2. Related Work

In terms of sensing tasks and the time and location of mobile users, there are many research works on task assignment and path planning to encourage mobile users to complete sensing tasks of high quality in PCS [8,22,23]. These studies mainly concern the task deadline instead of the duration needed to complete the task. Task assignment based on the available time of the mobile users and the location of tasks is considered in [24]. A two-stage task allocation framework is proposed, to optimize task allocation through a compromise between maximizing the total task quality and minimizing the total perceived time. Two distance comparison mechanisms are presented in [25]. These make use of geographical inseparability to make the users’ location privacy safer and minimize their travel distance. The tasks are allocated according to the location of users to improve the overall efficiency of location-based tasks by using the quality-aware online task assignment algorithm given in [26]. Sensitivity to the duration of each task and the ability of participants are considered to increase the number of tasks completed, as found in [27]. In order to assign tasks to participants, a two-stage task allocation framework is proposed by considering the time interval of task detection and execution given in [28]. The authors of [29] propose a multiple time-constrained task allocation problem in terms of semi-opportunistic mobile crowdsensing (SO-MTTA), with the goal of maximizing the sensing value obtained by the platform. The authors of [30] study the online task assignment problem in mobile crowdsensing, where each task has a specific time window for sensor data collection. The objective is to maximize the total profit of the platform over the whole sensing period. For location-dependent sensing tasks (LDSTs), when locations are farther away and the tasks yield low monetary rewards for the workers, these can be difficult to complete. In [31], the authors present a task-bundling reorganization mechanism (TBRM) to improve the platform utility of the MCS system.

There are many studies in the literature on the incentive mechanism [32,33,34,35,36]. An incentive mechanism for privacy protection and data quality awareness is proposed in [32]. Data quality can be quantified according to the deviation between reliable data and actual data. The users receive monetary rewards according to data quality. An incentive mechanism based on reverse auction and a fine-grained ability reputation system is put forward by the authors of [33]. The winner is selected via a greedy algorithm, and the reward is determined according to the user’s bid and fine-grained ability. A personalized task recommendation system is proposed in [34], which recommends tasks to users based on a recommendation score that comprehensively considers each user’s preferences and reliability. The authors of [35] propose a game-based incentive mechanism, named Incentive-G, aiming at recruiting mobile users effectively and improving the reliability and quality of sensing data against malicious users. The Incentive-G mechanism consists of several design phases, including analyzing the sensing data, determining the reputations of mobile users, and ensuring data quality and reliability by voting in a task group. This mechanism adopts a two-stage Stackelberg game for analyzing the reciprocal relationship between service providers and mobile users and then optimizes incentive benefits using backward induction. The authors of [36] propose a quality-driven online task-bundling-based incentive mechanism (QOTB). The design objective is to maximize social welfare while maximally satisfying the task quality requirements. QOTB introduces mental accounting theory to build accounts for task execution profit and bonus, respectively, which are then used to ascertain the participation willingness of workers. The paper adopts task bundling to stimulate workers to change their original travel schedules in order to balance task participation according to the popularity of task locations as well as the cost of travel. A comparison table listing the differences among the previous works is provided in Table 1.

## 3. System Model and Problem Statement

In this section, the system model of the PCS, followed by the problem statement, is presented. Table 2 shows the main notations used in this paper.

**Definition 1**  **(Task):***The requesters are asked to send tasks* $T=\{t_{1},t_{2},\ldots ,t_{n}\}$ *to the mobile crowdsensing platform, which are published dynamically upon their arrival. Each task* $t_{j}\in T$ *is associated with a task profile, which is represented by* $P_{j}=\{ec_{j},er_{j},pt_{j},gt_{j},E_{j}\}$*, encompassing the execution costs (*$ec_{j}$*), rewards (*$er_{j}^{i}$*), geographical position (*$pt_{j}$*), task grade (*$gt_{j}$*), and task characteristic set,* $E_{j}=\{e_{j}^{1},e_{j}^{2},\ldots ,e_{j}^{L}\}$*. In PCS, there are* L *characteristics. If* $t_{j}\in T$ *does not have the* k*-th execution characteristics, then* $e_{j}^{k}=0$.

**Definition 2**  **(Users):***The platform assigns tasks* $T=\{t_{1},t_{2},\ldots ,t_{n}\}$ *to mobile users* $U=\{u_{1},u_{2},\ldots ,u_{m}\}$ *without exceeding their time budget. Each worker* $u_{i}\in U$ *is associated with a specification, denoted by* $S_{i}=\{tb_{i},pu_{i},gu_{i},v_{i},\alpha_{i},A_{i}\}$*, which consists of the time budget (*$tb_{i}$*), current position (*$pu_{i}$*), user grade (*$gu_{i}$*), rate of travel (*$v_{i}$*), learning rate (*$\alpha_{i}$*), and the skill set of user* $A_{i}=\{a_{i}^{1},a_{i}^{2},\ldots ,a_{i}^{L}\}$. $a_{i}^{k}\in A_{i}$ *signifies that* $u_{i}\in U$ *possesses the* k*-th skill, corresponding to the* k*-th execution characteristic of task* $t_{j}\in T$*. In those instances where user* $u_{i}\in U$ *lacks the* k*-th skill,* $a_{i}^{k}=0$.

**Definition 3**  **(Users–Task Group):***The users–task group* $G_{j}^{i}=<u_{i},t_{j}>$ *denotes the assignment of a user* $u_{i}\in U$ *to carry out a task* $t_{j}\in T$*. During this process, the user expends time (denoted as* $time_{i}^{j}$*) on the task and accrues a profit (denoted as* $profit_{j}^{i}$*). The time is composed of two components: execution time (denoted as* $et_{j}^{i}$*) and journey time (denoted as* $jt_{j}^{i}$*), which is formulated as Equation (1). The profit, formulated as Equation (2), is calculated by subtracting the execution costs (*$ec_{j}^{i}$*) and journey costs (denoted as* $jc_{j}^{i}$*) from the execution rewards (*$er_{j}$*). The utility is defined as the ratio of each user’s profit to the time needed for completing the task, which is formulated as Equation (3).*


(1)
$$time_{i}^{j}=et_{j}^{i}+jt_{j}^{i}$$



(2)
$$profit_{j}^{i}=er_{j}^{i}-ec_{j}^{i}-jc_{j}^{i}$$



(3)
$$utility_{i}^{j}=\frac{profit_{j}^{i}}{time_{i}^{j}}=\frac{er_{j}^{i}-ec_{j}^{i}-jc_{j}^{i}}{et_{j}^{i}+jt_{j}^{i}}$$


*The execution time is usually influenced by the skill grade of the user and the experience of the user. The execution time is denoted as:*(4)$$et_{j}^{i}=\eta \sum_{k=0}^{L}\frac{e_{j}^{k}}{a_{i}^{k}+1}$$*where* η *is a time execution factor.* $a_{i}^{k}+1$ *is a precautionary measure designed to prevent an excessively large value when the user lacks skill* $a_{i}^{k}$ *or when the value* $a_{i}^{k}$ *of a is too small. This aligns with real-world scenarios where, in the absence of the necessary skills for a particular task attribute, users often resort to self-directed learning to acquire the required competencies.*


*The journey time and journey costs linearly escalate with distance, as expressed by Equations (5) and (6), respectively.*

(5)
$$jt_{j}^{i}=\frac{d(u_{i},t_{j})}{v_{i}}$$


(6)
$$jc_{j}^{i}=d(u_{i},t_{j})*\lambda$$



*The distance between the position of user* $u_{i}\in U$ *and the location of task* $t_{j}\in T$ *is denoted by* $d(u_{i},t_{j})$. λ *is a linear factor of the journey costs.*

**Definition 4**  **(Multi-task Assignment and Path-Planning Problem):***In the multi-task assignment and path-planning problem (MTAPP), each user* $u_{i}\in U$ *is assigned a task group* $G_{i}\subseteq T$*. In order to motivate users to complete more tasks in a unit of time, we define the utility* $utility_{j}^{i}$ *of* $u_{i}\in U$ *in terms of task assignment and path planning as the ratio of their profit to the time required to complete the task* $t_{j}\in G_{i}$*. Thus, the objective of the multi-task assignment problem is to minimize the utility while satisfying the execution requirements of each task and ensuring a rational allocation of tasks among users*.

(7)$$\max\sum_{U}\sum_{G_{i}}\frac{er_{j}-ec_{j}-jc_{j}^{i}}{et_{j}^{i}+jt_{j}^{i}}$$
s.t.
(8)$$\sum_{j=1}^{\left|G_{i}\right|}et_{j}^{i}+jt_{j}^{i}\leq tb_{i},\forall t_{j}\in G_{i}$$
(9)$$G_{i}\cap G_{s}=\emptyset ,\forall G_{i}\subseteq T,\forall G_{s}\subseteq T$$

Here, $\left|G_{i}\right|$ is the number of execution tasks that are set, $G_{i}$. Equation (8) indicates that the time $u_{i}\in U$ to complete the execution tasks $G_{i}\subseteq T$ should not exceed the time budget $tb_{i}$. Equation (9) ensures that each task is allocated to only one user.

## 4. Grade-Matching Degree and Similarity-Based Mechanism

This section provides a detailed introduction and discussion of the grade-matching degree and similarity-based mechanism (GSBM). The overall framework of the GSBM is described in Figure 1 and comprises two main phases:(1)Preprocessing phase: In this phase, the mobile crowdsensing platform divides the arriving tasks and users by region. At this stage, the mobile crowdsensing platform provides tasks to users according to region. Subsequently, the data are preprocessed, introducing the concepts of similarity and grade-matching degree. The grade-matching degree pricing mechanism is then introduced, where the matching degree determines a portion of the expected reward. Users can only obtain the full amount of the expected rewards by completing tasks corresponding to their grade. A mathematical model based on similarity is constructed to delve into the impact of task similarity on user task completion, thereby incentivizing users to engage more in those tasks matching their abilities.(2)Determining tasks phase: In this phase, to address the MTAPP problem, an improved ant colony optimization (IACO) algorithm is employed to maximize the overall utility of users during task execution.

### 4.1. Grade-Matching Degree Pricing 

**Definition 5**  **(Grade-Matching Degree):***The grade-matching degree between the tasks and users is measured by the Euclidean distance. The grade-matching degree between the user* $u_{i}\in U$ *and the tasks* $t_{j}\in T$ *is denoted as:*(10)$$match(u_{i},t_{j})=\frac{1}{1+\sqrt{{\left(gu_{i}-gt_{j}\right)}^{2}}}$$*When the user ability grade equals the task difficulty grade, the grade-matching degree* $match(u_{i},t_{j})=1$. *Otherwise,* $0<match(u_{i},t_{j})<1$.*In order to encourage users to complete those tasks of the corresponding grades, the value of the grade-matching degree determines the proportion of the expected rewards. The earnings of the user* $u_{i}\in U$ *performing the task* $t_{j}\in T$ *are denoted as follows:*(11)$$er_{j}^{i}=er_{j}*match(u_{i},t_{j})$$

### 4.2. Task Similarity Model

**Definition 6**  **(Task Similarity):***For a given two tasks, the similarity between tasks is defined as the relationship among their respective attributes. A measurement method utilizing the Pearson correlation coefficient [37] is employed to depict the similarity between tasks. The similarity between tasks*   $t_{j}\in T$ *and* $t_{g}\in T$ *is determined as follows:*
(12)$$sim(t_{j},t_{g})=\frac{\sum_{k=1}^{L}\left(e_{j}^{k}-{\bar{E}}_{j})(e_{g}^{k}-{\bar{E}}_{g}\right)}{\sqrt{\sum_{k=1}^{L}{\left(e_{j}^{k}-{\bar{E}}_{j}\right)}^{2}}\sqrt{\sum_{k=1}^{L}{\left(e_{g}^{k}-{\bar{E}}_{g}\right)}^{2}}}$$*where* ${\bar{E}}_{j}$ *and* ${\bar{E}}_{g}$ *are the mean values of* $E_{j}$ *and* $E_{g}$
*, respectively. In Equation (9), the values obtained through the Pearson correlation coefficient measurement range from 0 to 1, as shown by* $sim(t_{j},t_{g})\in [0,1]$
*. A value closer to 1 indicates a stronger linear relationship between two tasks, implying a higher degree of similarity.*

To elucidate the influence of similarity on the execution time of tasks performed during user engagement, the mechanism is grounded in an S-shaped learning curve [24] model, formulating a learning curve mathematical model that relies on task similarity. In this model, only the influence of the preceding task on the current task is taken into account, while disregarding the potential impact of other preceding tasks, as illustrated below.
(13)$$et_{j}^{i}=\eta \sum_{k=0}^{L}\frac{e_{j}^{k}}{a_{i}^{k}+1}{\left(1+sim(t_{j},t_{j-1})\right)}^{\theta_{i}}$$
(14)$$\theta_{i}=\ln(1-\alpha_{i})/\ln2$$

Task $t_{j-1}\in G_{i}(j\neq 1)$ is the front task of task $t_{j}\in G_{i}$. $\theta_{i}$, as denoted by Equation (14), is the learning index ($\theta_{i}<0$). 

### 4.3. IACO Algorithm

The MTAPP solution space is very large, and traditional combinatorial optimization algorithms are inefficient at handling the multitask allocation problem. In this section, we introduce the IACO algorithm to solve the multitask allocation problem. 

In this section, we introduce an enhanced iterative ant colony optimization (IACO) algorithm, as presented in Algorithm 1, which integrates the principles of an ant colony optimization (ACO) algorithm [24,38] and a greedy algorithm. The ACO algorithm is a heuristic optimization algorithm inspired by foraging behavior and information exchange among ants. When ants discover food, they return to the colony and leave pheromone trails, guiding other ants to follow the same path. The fading of these pheromones over time prevents the algorithm from converging to a local optimum. Conversely, the greedy algorithm relies on local optimal choices at each step, aiming to achieve a globally optimal solution through successive steps. This combination enhances the exploration–exploitation balance for improved optimization.
**Algorithm 1**. IACO algorithmSet parameters: ant (number of all ants), ant (number of iteration), α, β, ρ, Q Input: T, U Output: best tasks path list G Initialize heuristic information and pheromone matrices. 1. for i=1:iter do 2.  for j=1:ant do 3.   for u=1:m do 4.    Calculate the utility between users and tasks; 5.    Calculate selection probability; 6.    if i=1
7.     Select next task with the greatest utility 8.    else 9.     Select next task by roulette; 10.    end if 11.    Add the selected task into taboo list; 12.    Leave the pheromone 13.   end for 14.  end for 15. end for 16. return taboo list

In the IACO algorithm, each ant maintains a taboo list in order to store the tasks that are executed sequentially. Each taboo list represents a solution to the MTAPP problem. The first ant is compelled to follow the route determined by the greedy algorithm. In other words, one ant selects that task with the maximum utility as the next task. Other ants choose the next task based on the transition probabilities at each step of path construction. Each ant moves along the chosen path, updating the concentration of pheromones on the selected route. After all ants have completed their movement, the pheromones evaporate to simulate the forgetting of information in reality. Subsequently, pheromones accumulate on the ant paths to enhance the information trail. This process of ant movement and pheromone update is repeated until the maximum iteration count is reached. The resulting path list represents the solution of MTAPP.

In the IACO algorithm, the probability $Pro_{i}^{j}$ is calculated by heuristic information $utility_{i}^{j}$ and pheromone $\tau^{j}$, as shown in Equation (15).
(15)$$Pro_{i}^{j}={\left(utility_{i}^{j}\right)}^{\alpha }\times {\left(\tau^{j}\right)}^{\beta }$$
(16)$$\tau^{j}=(1-\rho )\cdot \tau^{j}+\Delta \tau^{j}$$
(17)$$\Delta \tau^{j}=\sum_{k=1}^{ant}\Delta \tau_{k}^{j}$$
(18)$$\Delta \tau_{k}^{j}=\left\{\begin{array}{cc} Q & \mathrm{ant}\;k\;\mathrm{go}\;\mathrm{through}\;\mathrm{the}\;\mathrm{task}\;t_{j}\\ 0 & \mathrm{ant}\;k\;\mathrm{doesn}’t\;\mathrm{go}\;\mathrm{through}\;\mathrm{the}\;\mathrm{task}\;t_{j} \end{array}\right.$$

α, β represents the weight factor. In Equation (16), the pheromone $\tau^{j}$ is divided into two parts. $(1-\rho )\cdot \tau^{j}$ represents the assumption that the pheromones on all tasks will evaporate after each search. ρ is the evaporation coefficient of the pheromone on the tasks in the taboo list. $\Delta \tau^{j}$ represents the increment of pheromones in each iteration, which is defined in Equation (17), where $\Delta \tau_{k}^{j}$ represents the increment of pheromone so that the k-th ant stays in task in this iteration, as shown in Equation (18). The constant coefficient Q is taken as the pheromone.

## 5. Evaluation

### 5.1. Experimental Settings

In this group of experiments, mobile users and perception tasks are randomly placed in a unit area of 5×5 square kilometers. The mobile user has 3 skills, with attribute values of [0, 50], and the perception task has 3 attributes, with attribute values of [0, 100]. We have categorized the tasks and users into 3 grades. The user’s budget time is set to 1–8 h. The expected rewards of the tasks are set at 10–40. The execution costs of the tasks are set at 5–20.

The other experimental settings are shown in Table 3.

### 5.2. Performance Metric

Referring to the experiments in Reference [1], simulations were conducted by varying two parameters to simulate different MTAPP scenarios, including the number of tasks and the number of users. The evaluation of the method’s performance was based on studying how various parameters affected the following indicators.

(1)Task coverage rate: This metric measures the ratio of completed tasks to the total number of tasks, whereby a higher coverage rate indicates the algorithm’s comprehensiveness and efficiency in terms of task assignment.(2)Average task completion rate: Calculated as the number of tasks completed within a unit of time, this metric reflects the efficiency and timeliness of task execution.(3)User profits: This metric assesses the impact of task assignment on user profits, ensuring that the algorithm considers the economic interests of users while optimizing task allocation.(4)Rationality of task assignment: This metric evaluates whether the system can reasonably assign tasks to suitable users, considering the ratio of tasks completed by users with comparable abilities to the total tasks completed by all users. A higher ratio indicates better performance of the mechanism.

### 5.3. Baseline Approaches

Location and task-characteristics-based task allocation (LTCTA): In [39], LTCTA is employed to allocate a set of tasks to a group of users and generate a sequence of paths list. LTCTA assesses the rationality of the allocation by considering the geographical information and task features, specifically weighing route distance, task similarity, and task priority. The weighted results serve as the optimization objective. A greedy algorithm is employed for task assignment. 

A worker multitask allocation–genetic algorithm (WMTA-GA): In [1], a WMTA-GA is designed to assist workers in selecting multiple tasks while considering both the time constraints of workers and the requirements of the tasks. Additionally, a pricing mechanism is employed to determine the budget for each task, and then the workers’ wages are determined based on willingness factors. To address this issue, a genetic algorithm is proposed to maximize worker welfare.

### 5.4. Result Analysis

#### 5.4.1. Algorithm Performance Analysis

By examining Figure 2, we can assess the performance of different algorithms under varying user numbers. Notably, in Figure 2a, the GSBM demonstrates a superior task coverage rate compared to the LTCTA and WMTA-GA. The limited tasks achievable by all users within a given time budget are apparent when the user numbers are small. As the user number increases, the potential task quantity also rises, resulting in an enhanced task coverage rate. When the number of tasks is greater than 300, the task coverage rate in the GSBM reaches 1. Turning our attention to Figure 2b, it is evident that the GSBM exhibits the highest performance in terms of the average task completion rate, and this rate increases proportionally with the number of users. Specifically, within the unit time, the GSBM achieves the completion of two to three tasks, whereas the LTCTA and WMTA-GA only accomplish 0.5 to 1 task. This performance disparity can be attributed to the GSBM’s consideration of the impact of similarity on task execution time. In the GSBM, users opt for tasks with high similarity, consequently reducing the task execution time. Consequently, under identical time constraints, users utilizing the GSBM complete more tasks relative to the LTCTA and WMTA-GA. The GSBM significantly improves both the task coverage rate and average time coverage rate.

In Figure 2c, as the number of users increases, user profits gradually rise. However, when the user number reaches 400, user profits no longer exhibit a significant change with further increases in user number. This is because, at this point, the number of users is sufficient to complete all tasks. The LTCTA consistently shows the lowest performance in user profits, consistently lagging behind the GSBM. This can be attributed to the fact that the LTCTA does not consider the profits of the users in task execution. It is noteworthy that in scenarios with a smaller number of users, the GSBM outperforms the WMTA-GA in terms of user profits. However, when the user number reaches 400, the GSBM’s user profits are slightly lower than those of the WMTA-GA by 200. This difference arises from the fact that the WMTA-GA optimizes for user profits, and the WMTA-GA does not incorporate a grade-matching degree pricing mechanism. Consequently, in the WMTA-GA, the users receive full rewards regardless of the level of tasks completed. 

As shown in Figure 2d, the WMTA-GA exhibits relatively poor rationality in task allocation, falling below 0.5. This can be attributed to the WMTA-GA completely neglecting the impact of matching user capabilities with task difficulty on task assignment. In contrast, both the GSBM and LTCTA demonstrate high performance in the rationality of task assignment, with the GSBM slightly outperforming the LTCTA and gradually approaching 0.8 with an increasing number of users. In the LTCTA, the consideration of task priority during task execution is evident. Therefore, in our simulation, we treat grade-matching degree as equivalent to task priority in the LTCTA, weighted along with similarity as an optimization objective. In the GSBM, the influence of the grade-matching degree and similarity is quantified, with utility serving as the optimization objective. Consequently, the GSBM excels in achieving an outstanding assignment between user capabilities and task difficulty.

By examining Figure 3, we can evaluate the performance of different algorithms under varying task numbers. In Figure 3a, as the task number increases, the task coverage gradually decreases. This is because, with an increasing number of tasks, the limitations of resources and workers may prevent the efficient completion of all tasks, thereby reducing the overall task coverage. Clearly, the GSBM exhibits higher task coverage compared to the LTCTA and WMTA-GA.

In Figure 3c, user profits with varying task numbers for the different algorithms are illustrated. As the number of users increases, user profits gradually increase. However, upon reaching a task number of 400, user profits cease to significantly change along with further increases in task number. This phenomenon arises because, upon surpassing a certain task quantity, this exceeds the limits of the user capabilities, resulting in the stabilization of user profits. In this trend, the LTCTA consistently exhibits the lowest performance in terms of user profits, consistently falling behind the GSBM. When the task number is low, the GSBM surpasses the WMTA-GA in user profits. However, with an increase in task number, the GSBM’s user profits are slightly lower than those of the WMTA-GA, which optimizes user profits. This suggests that the GSBM demonstrates outstanding performance compared to the WMTA-GA in terms of user profits, particularly as task numbers increase.

As depicted in Figure 3d, the GSBM consistently exhibits superior task assignment rationality, maintaining a high level between 0.7 and 0.8. In contrast, the WMTA-GA shows relatively poor rationality in task assignment, consistently scoring below 0.5. The LTCTA’s task assignment rationality excels when the task quantity is low but gradually decreases as the task number increases, ultimately falling below 0.5. This suggests that the task scale has a limited impact on the GSBM’s task assignment rationality, while the LTCTA is not suitable for large-scale task assignments.

#### 5.4.2. Individual Rational Analysis

In terms of algorithm performance, the GSBM ensures individual rationality for both users and tasks [36]. As depicted in Figure 4a, the final earnings for each user are not lower than the cost of completing their tasks, ensuring a non-negative utility for each candidate user. As shown in Figure 4b, the ultimate task allocation results provide the budget for each task and the cost of payment to the users. At the conclusion of the algorithm, the total cost of payment for a task does not exceed its budget, ensuring a non-negative utility for each task. Throughout the task allocation process, the overall payment curve for tasks follows the same trend as the budget curve, wherein user payments always exceed the cost of executing their tasks. The trend of the payment curve aligns with the trend of the task execution cost curve.

## 6. Conclusions

This paper proposes a multi-task assignment and path-planning problem (MTAPP). In the MATPP, the ratio of the profit gained by users to the time spent by users to complete tasks is taken as the utility of users to complete tasks, which is the standard consideration in task assignment and path planning. A grade-matching degree and task similarity-based mechanism (GSBM) is proposed to solve the MATPP. Finally, the IACO algorithm is used to maximize the utility. A large number of experiments show that the GSBM achieves good results. 

In the GSBM, grade matching between users and tasks is considered, but matching between the skill level of the users and the attributes of the tasks is not considered. We plan to design a fine-grained matching scheme in the future to realize the matching of skills and attributes between users and tasks.

## Figures and Tables

**Figure 1 sensors-24-00651-f001:**
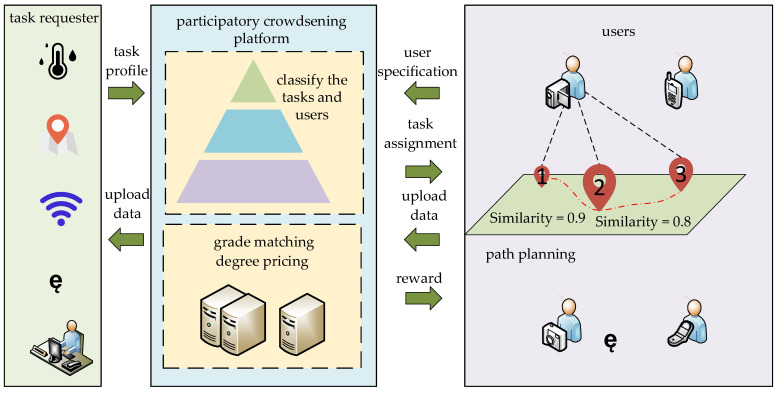
PCS system.

**Figure 2 sensors-24-00651-f002:**
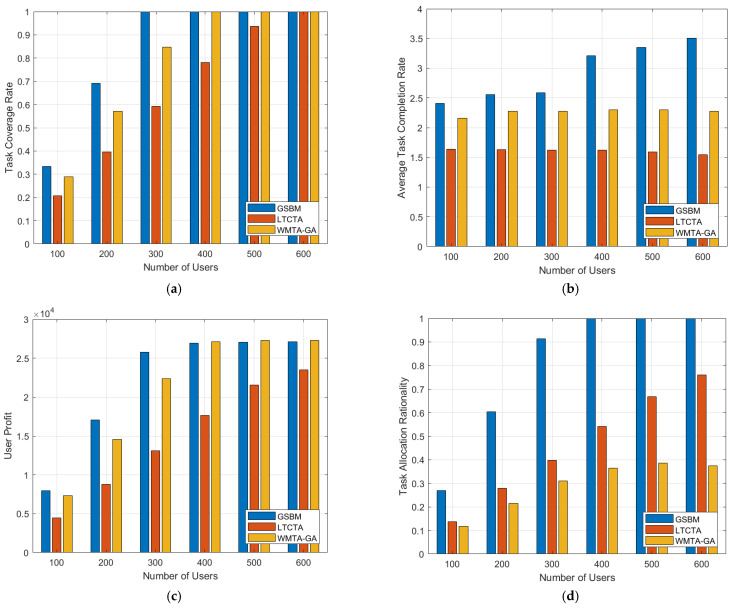
Performance comparison chart according to different numbers of users. (**a**) Task coverage rate; (**b**) Average task completion rate; (**c**) User profit; (**d**) Task allocation rationality.

**Figure 3 sensors-24-00651-f003:**
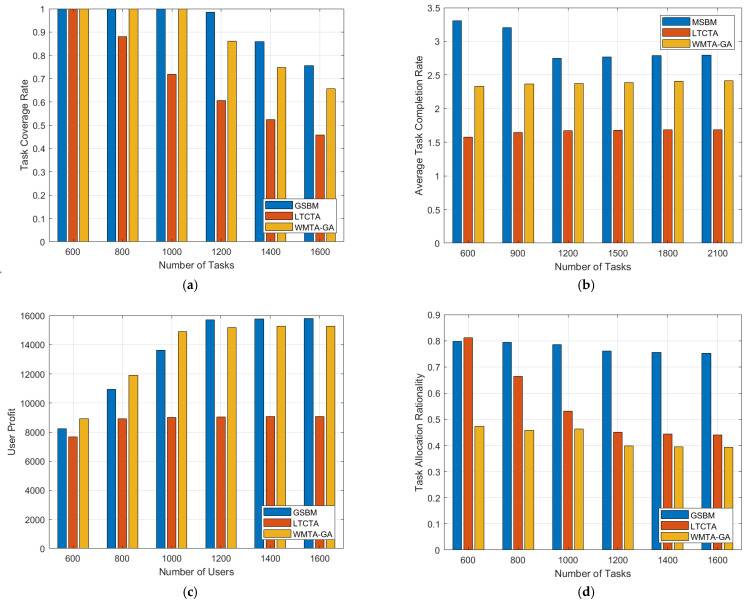
Performance comparison chart according to different numbers of tasks. (**a**) Task coverage rate; (**b**) Average task completion rate; (**c**) User profit; (**d**) Task allocation rationality. In Figure 3b, the GSBM demonstrates the highest performance in terms of average task completion rate. As the task quantity increases, the average task completion rate gradually decreases, eventually stabilizing. This is due to the rising task number, leading to difficulties in maintaining efficient task execution due to the limitations of worker capabilities, resulting in a decline in the average task completion rate. Ultimately, the system reaches a balanced state, and the task completion rate stabilizes, reflecting the limit of the number of tasks that the system can effectively handle under the given conditions.

**Figure 4 sensors-24-00651-f004:**
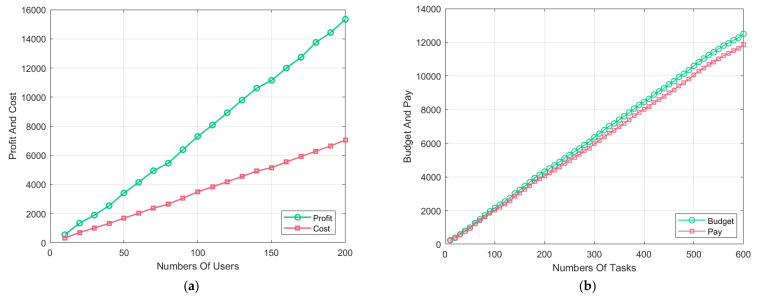
Individual rationality of the users and tasks. (**a**) Individual rationality of the users; (**b**) Individual rationality of the tasks.

**Table 1 sensors-24-00651-t001:** The differences among the previous works.

Comparison Table	
[24]	minimize aggregate sensing time and maximize total task quality
[25]	maximize the total task quality and minimize the total perceived time
[26]	optimize task quality, based on location
[27]	maximize the number of completed tasks under the constraints of sensing duration and task capacity of each worker
[28]	maximize the overall sensing type matching degree and POI coverage
[29]	maximize the sensing value under the time constraint
[30]	maximize the total profit in the whole sensing period within the time window
[31]	improve the platform utility through a task bundling reorganized mechanism (TBRM)
[32]	motivate the participation of task participants by providing appropriate monetary rewards
[33]	maximize the social cost to satisfy the fine-grained ability requirement
[34]	propose a personalized task recommender system that jointly takes each user’s preference and reliability into consideration
[35]	improve the reliability and quality of sensing data by the Incentive-G mechanism
[36]	maximize social welfare while maximally satisfying the task quality requirements

**Table 2 sensors-24-00651-t002:** The main notations used in this paper.

System Model Parameter	
Task Parameter	
T	Set of all tasks
n	Number of tasks
$t_{j}$	Task j
$P_{j}$	Task profile of task j
$ec_{j}$	Execution costs of task j
$er_{j}$	Expected rewards of task j
$pt_{j}$	Geographical position of task j
$gt_{j}$	Grade of task j
$E_{j}$	Characteristic set of task j
$e_{j}^{k}$	The k-th characteristics of task j
**User Parameter**	
U	Set of all users
m	Number of users
$u_{i}$	User i
$S_{i}$	User specification of user i
$tb_{i}$	Time budget of task j
$pu_{i}$	Current position of task j
$gu_{i}$	Grade of task j
$v_{i}$	Travel rate of task j
$\alpha_{i}$	Learning rate of task j
$A_{i}$	Skill set of task j
$a_{i}^{k}$	The k-th skill of task j
$G_{i}$	The assignment and path list of a user: $u_{i}\in U$
**Users-Task Group**	
$G_{j}^{i}$	The assignment of a user $u_{i}\in U$ to carry a task $t_{j}\in T$
$time_{i}^{j}$	Total time of a user $u_{i}\in U$ to carry a task $t_{j}\in T$
$profit_{j}^{i}$	Profit of a user $u_{i}\in U$ to carry a task $t_{j}\in T$
$utility_{i}^{j}$	Utility of a user $u_{i}\in U$ to carry a task $t_{j}\in T$
$et_{j}^{i}$	Execution time of a user $u_{i}\in U$ to carry a task $t_{j}\in T$
$jt_{j}^{i}$	Journey time of a user $u_{i}\in U$ to carry a task $t_{j}\in T$
$jc_{j}^{i}$	Journey costs of a user $u_{i}\in U$ to carry a task $t_{j}\in T$
$er_{j}^{i}$	Rewards of a user $u_{i}\in U$ to carry a task $t_{j}\in T$
$ec_{j}^{i}$	Execution costs of a user $u_{i}\in U$ to carry a task $t_{j}\in T$
$d(u_{i},t_{j})$	Distance of a user $u_{i}\in U$ to carry a task $t_{j}\in T$
$match(u_{i},t_{j})$	Grade-matching degree between a user $u_{i}\in U$ and a task $t_{j}\in T$
$sim(t_{j},t_{g})$	Task similarity between a task $t_{g}\in T$ and a task $t_{j}\in T$

**Table 3 sensors-24-00651-t003:** The experimental settings.

Parameter	Value	Description
K	3	Grades of users and tasks
$\omega_{k}$	0.3, 0.3, 0.4	Weight of the attributes
$\delta_{k}$	0.3, 0.3, 0.4	Weight of the skills
λ	3	Linear factor of toll
η	800	Execution time factor
$v_{i}$	25	Speed of users
$\alpha_{i}$	0.5	Learning rate of user
ant_num	150	Ant population
iter_max	100	Maximum number of iterations
α	1	Pheromone weighting factor
β	5	Heuristic function weighting factor
Q	1	Constant coefficient
ρ	0.2	Pheromone volatilization factor

## Data Availability

Data are contained within the article.

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
