# Peer review of "Task Assignment and Path Planning Mechanism Based on Grade-Matching Degree and Task Similarity in Participatory Crowdsensing"

_sensors, 2024, doi:10.3390/s24020651_

Round 1

Reviewer 1 Report

Comments and Suggestions for Authors

 In this paper, a grade similarity-based mechanism (GSBM) is proposed. The mechanism quantifies task similarity and its impact on time considering that performing similar tasks by users reduces the time cost of task execution. In addition, the mechanism proposed grade matching degree to solve the problem of users preferring simple tasks. Chi-squared binning were used to grade sensing tasks and mobile users. Grade matching degree that measured by the Euclidean distance determines the proportion of rewards received by mobile users. The ratio of profit to time performing tasks is defined as the utility of users performing tasks, which is used to describe users' preferences for different tasks. Finally, the improved ant colony optimization(IACO) algorithm which combines ant colony algorithm and greedy algorithm is adopted to maximize total utility. The simulation results show that GSBM can increase the number of tasks completed and the rewards obtained by users at the same time cost, while also improving the performance and completion rate of difficult tasks.

The topic is interesting, however, paper organization should be improved. Extensive revision is required. Detailed comments are given as:

1. The motivation in abstract and introduction should be improved. In addition, research gap should be clearly mentioned in abstract and introduction.

2. It is suggested to provide a suitable reference for the first line in introduction. The following paper maybe helpful:  https://doi.org/10.3390/s22083013

3. Authors should add some more recent works in related work section, e.g., https://doi.org/10.3390/s22083013. Also provide a comparison table listing the differences among previous works and the proposed work.

4.  Please explain the importance of the proposed solution. In addition, please clearly describe the main contributions of the manuscript in details.  

5. It is suggested to provide the main  pseudocode of the proposed solution and discuss it step by step. In addition, what are the advantages and disadvantages of the proposed approach?

6. In result section, authors should discuss each table and figure in details, also discuss why their proposed solution performs better than other solutions by providing suitable results. Please provide high quality HD figures.

7. Please check the whole manuscript for typos and grammar errors, and improve the language of the paper. 

Comments on the Quality of English Language

 Extensive editing of English language required

Reviewer 2 Report

Comments and Suggestions for Authors

1. The author must mention, which type of crowd sensing is used in the proposed method( Participatory crowdsensing / Opportunistic crowdsensing)

2. There is a huge lag in continuity between one concept with the other( Readability is very hard)

3. Too many concepts without exploring much like Chi-squared binning, IACO, GSBM............

4. Figures from 2 to 7 - Presentation quality is very poor, results and discussion section for each figure must be improved and highlighted and correlation  between sections must be highlighted

5. Paper concept is good but very to understand in the present form, needs more correlation and explanation between sections

6. Performance comparison wit recent literature articles must be done and justify how the proposed scheme is novel?

7. How Mobile crowdsensing  is carried out? Explain the stages along with the data in detail

8. Requires proof reading

9. How IACO plans for optimal path, explain in detail

10.How the the level of difficulty for each task is assessed?

11. No metrics have been evaluated through out the paper, for every scheme utilized there are n number of validation parameters available, use them wisely

Comments on the Quality of English Language

Needs proof reading completely

Round 2

Reviewer 1 Report

Comments and Suggestions for Authors

Authors have addressed all my concerns in the revised version, it can be accepted for publication

Comments on the Quality of English Language

Minor editing is required